Identification of a three-gene-based prognostic model in multiple myeloma using bioinformatics analysis

Pan Ying
Meng Ye
Zhai Zhimin zzzm889@163.com
Xiong Shudao xshdao@ahmu.edu.cn
Department of Hematology, The Second Affiliated Hospital of Anhui Medical University , Hefei , China
Juan Hsueh-Fen
Electronic publication date: 2021 Jun 28
Publication date: 2021
Volume: 9
Electronic Location ID: e11320
Received 2020 Nov 12; Accepted 2021 Mar 31
Copyright: © 2021 Pan et al.
Copyright year: 2021
Copyright holder: Pan et al.
License: This is an open access article distributed under the terms of the Creative Commons Attribution License, which permits unrestricted use, distribution, reproduction and adaptation in any medium and for any purpose provided that it is properly attributed. For attribution, the original author(s), title, publication source (PeerJ) and either DOI or URL of the article must be cited.
License URL: https://creativecommons.org/licenses/by/4.0/

Keywords: Multiple myeloma, Prognosis, WGCNA, Bioinformatics analysis, Prognostic model

Funding: China No. 81670179 Second Hospital of Anhui Medical University 2019GQFY11 and 2018KA09 This work was supported by the National Natural Science Foundation of China (No. 81670179), the National Natural Science Foundation Incubation Project of the Second Hospital of Anhui Medical University (No. 2019GQFY11) and the 4th Science and Technology New Star Training Program of the Second Hospital of Anhui Medical University (No. 2018KA09). The funders had no role in study design, data collection and analysis, decision to publish, or preparation of the manuscript.

==============================
Background

Multiple myeloma (MM), the second most hematological malignancy, has high incidence and remains incurable till now. The pathogenesis of MM is poorly understood. This study aimed to identify novel prognostic model for MM on gene expression profiles.

Methods

Gene expression datas of MM (GSE6477, GSE136337) were downloaded from Gene Expression Omnibus (GEO) database. The differentially expressed genes (DEGs) in GSE6477 between case samples and normal control samples were screened by the limma package. Meanwhile, enrichment analysis was conducted, and a protein-protein interaction (PPI) network of these DEGs was established by STRING and cytoscape software. Co-expression modules of genes were built by Weighted Correlation Network Analysis (WGCNA). Key genes were identified both from hub genes and the DEGs. Univariate and multivariate Cox congression were performed to screen independent prognostic genes to construct a predictive model. The predictive power of the model was evaluated by Kaplan–Meier curve and time-dependent receiver operating characteristic (ROC) curves. Finally, univariate and multivariate Cox regression analyse were used to investigate whether the prognostic model could be independent of other clinical parameters.

Results

GSE6477, including 101 case and 15 normal control, were screened as the datasets. A total of 178 DEGs were identified, including 59 up-regulated and 119 down-regulated genes. In WGCNA analysis, module black and module purple were the most relevant modules with cancer traits, and 92 hub genes in these two modules were selected for further analysis. Next, 47 genes were chosen both from the DEGs and hub genes as key genes. Three genes (LYVE1, RNASE1, and RNASE2) were finally screened by univariate and multivariate Cox regression analyses and used to construct a risk model. In addition, the three-gene prognostic model revealed independent and accurate prognostic capacity in relation to other clinical parameters for MM patients.

Conclusion

In summary, we identified and constructed a three-gene-based prognostic model that could be used to predict overall survival of MM patients.

Introduction

Multiple myeloma (MM) is the second most common hematological malignancy, and caused by abnormal monoclonal plasma cells. It accounts for 10% of all hematological malignancies (Mateos & San Miguel, 2017). Many novel therapy strategies have been developed, and the survival of MM patients has improved considerably (Nooka et al., 2019; Ocio et al., 2014). However, the prognosis remains poor and the disease is still incurable (Nijhof et al., 2018). It is of great significance to identify novel cancer-specific prognostic biomarkers and construct prognostic model to find high-risk MM patients.

The prognostic models of MM usually include clinical information and cytogenetic abnormalities (Qi et al., 2020). International Staging System (ISS) and the revised ISS (R-ISS) are the most commonly used prognostic models. Recently, some prognostic models based on gene expression profile have been used to predict prognosis in MM, and even better than R-ISS (Liu et al., 2019; Bai et al., 2020). Besides, new technologies and methods such as single cell RNA-Seq (scRNA-Seq) have been used to predict the prognosis of MM (Jang et al., 2019). Weighted gene co-expression network (WGCNA) is also a novel method which uses numerous data in public databases to explore the underlying mechanism of gene networks, and usually used for gene prediction in large-scale datasets in recent years (Zhang & Horvath, 2005; Langfelder & Horvath, 2008). It first divides gene co-expression networks of complicated biological processes into several modules, then analyze their potential association with clinical traits to look for key modules and hub genes (Zhang & Horvath, 2005). It has been widely utilized in many malignant diseases as well as non-malignant diseases (Li et al., 2020b; Xu et al., 2020). However, it is rarely used in MM.

In the present study, we aimed to use WGCNA method to identify co-expression networks associated with MM and to find potential prognostic biomarkers and construct prognostic model for MM patients on gene expression profiles.

Materials and Methods

Data preparation

The raw data of GSE6477 and GSE136337 were downloaded from the gene expression omnibus (GEO) website (http://www.ncbi.nlm.nlh.gov/geo/). There were 101 case and 15 normal control in GSE6477, was based on GPL96Platforms ([HG-U133A] Affymetrix Human Genome U133A Array). There were 401 cases with overall survival (OS) and other clinical information was chosen to identify prognostic genes and construct the prognostic model in GSE136337. All the patients were randomly divided into 2 groups (60% as the training set and 40% as the test set). The Robust multi-array average (RMA) method was used to normalize data.

Analysis of differentially expressed genes (DEGs)

Limma package was used to identify differentially expressed genes (DEGs) (Ritchie et al., 2015), and the threshod value was |log2foldchange| >1.0 and adj.P.Val <0.05.

GO term and KEGG pathway enrichment analysis

To better explore the biological significance of DEGs, Gene Ontology (GO) and Kyoto Encyclopedia of Genes and Genomes (KEGG) pathway enrichment analysis were performed (Yu et al., 2012). q value <0.05 was used as the cutoff value.

A protein-protein interaction (PPI) network

A PPI network of these DEGs was constructed by using the Search Tool for the Retrieval of Interacting Genes (STRING) database (https://string-db.org/) (Szklarczyk et al., 2015).

Co-expression network analysis

The co-expression network was constructed by WGCNA to explore the correlation of genes (Top50%mad) and search for important interacted gene modules. PickSoftThreshold function of the WGCNA was used to calculate the soft thresholding power β value, and the β value was set as 6. Then, we constructed the co-expression modules using WGCNA algorithms with R software package. Next, Gene significance (GS) and Module membership measures were used to identify genes with high clinical significance as well as high Module membership. Module membership was the correlation between gene expression profile and module eigengene. While, GS was the absolute value of the correlation between gene expression and module traits.

Identification of prognostic key genes and construction of the prognostic model

Key genes were identified both from hub genes and the DEGs (Xia et al., 2018). All the patients were randomly divided into two groups (60% as the training set and 40% as the test set) and univariate and multivariate Cox regression analyses were used to investigate the correlation between OS and the expression level of key genes. P-value <0.05 was considered of significant difference in the univariate analysis. Key genes with P < 0.05 based on the univariate analysis were further included in the multivariate Cox regression analysis. Finally, a prognostic gene signature of MM patients was established based on a linear combination of the regression coefficient derived from the multivariate Cox regression model (β) multiplied with its expression level. The patients were separated into low- and high-risk groups based on the optimal cut-off value (Camp, Dolled-Filhart & Rimm, 2004) Kaplan–Meier (K–M) survival curves, time-dependent receiver operating characteristic (ROC) curve analyses and area under the curve (AUC) were used to evaluate the prediction accuracy of our model (Heagerty & Zheng, 2005). We also validated the prognostic model in the test set and whole set by K–M curves, ROC curve analyses, and AUC.

Identification of independent prognostic parameters of MM

To investigate independent prognostic parameters and to identify whether the prognostic model was the independent prognostic parameter, univariate and multivariate Cox regression analyses were performed using the Cox regression model method with forwarding stepwise procedure. Other clinical parameters including age, gender, number of transplantation (NOT), tumor staging (ISS and R-ISS), cytogenetic abnormalities, lactate dehydrogenase (LDH) level, albumin level, β2-microglobulin level, and percentage of abnormal plasma cell in bone marrow. P-value <0.05 was considered of significant difference. Also, we compared the AUC of our model with other conventional models such as age, gender, ISS, and R-ISS models.

Results

Identification of DEGs in MM and the enrichment of these DEGs

The DEGs of GSE6477 were screened by the limma package. A total of 178 DEGs were identified, including 59 upregulated genes and 119 downregulated genes in MM case samples compared to normal control samples (Fig. 1A). The DEGs were shown on a heatmap (Fig. 1B). GO analysis showed that these genes were significantly enriched in neutrophil activation, humoral immune response, and structural constituent of ribosome (Fig. 1C). In KEGG analysis, the DEGs were mainly enriched in Asthma, Staphylococcus aureus infection, and Ribosome (Fig. 1D). Next, A PPI network, as demonstrated in Fig. 2, was constructed using the online STRING website to understand the interaction among the related DEGs in MM.

Figure 1 The DEGs identified in GSE6477.

(A) Volcano map of DEGs between MM case group and normal control group. (B) Heatmap of the DEGs. (C) The top 20 GO terms in the enrichment analysis of DEGs. (D) The top 30 KEGG pathways in the enrichment analysis of DEGs.

Figure 2 Analysis of the PPI network.

A total of 178 DEGs were filtered into the DEGs PPI network complex.

Co-expression analysis and hub genes identification

WGCNA was used to construct co-expression network and to find the modules of highly correlated genes. Firstly, we calculated the soft thresholding power β, and the power of β was set at 6 to ensure a scale-free network in our research (Figs. 3A–3D). Next, we identified 15 modules for further analysis (Fig. 4A). The heatmap and meta-modules were used to visualize the gene network (Figs. 4B and 4C). Among these 15 modules, module black and module purple were the most relevant modules with cancer traits (Fig. 4D). Subsequently, intramodular analysis of the genes in the 15 modules was followed. And genes in the black and purple modules were found to have high correlations with MM (Figs. 4E and 4F). Finally, genes in these two modules were selected as hub genes according to the cut-off criteria: module membership values >0.80 and GS values >0.2. Totally, there were 92 hub genes from black and purple modules that were chosen for next analysis.

Figure 3 Determination of soft thresholding power.

(A) Analysis of the scale-free fitting indices for various soft thresholding powers. (B) Mean connectivity analysis of various soft thresholding powers. (C) Histogram of the connection distribution when β = 6. (D) Check scale-free topology when β = 6.

Figure 4 Identification of modules associated with MM clinical traits.

(A) Clustering dendrograms of genes based on co-expression network analysis. In total, 15 co-expression modules were constructed and shown in different colors. (B) A heatmap of all the genes. The light color indicated a low overlap, and the darker red color indicated higher overlap. (C) The eigengene dendrogram and eigengene adjacency plot. (D) Module-trait association. The purple and black gene modules were the most relevant modules with cancer traits. (E) A scatter plot of gene significance for MM versus the module membership in the black module. (F) A scatter plot of gene significance for MM versus the module membership in the purple module.

Identification of survival related key genes and construction of three-gene prognostic signature

Key genes were identified both from hub genes and the DEGs. Totally, 47 key genes were identified (Fig. 5). To evaluate the prognostic values of these key genes in MM patients, we next performed a univariate Cox regression, and identified 8 genes significantly related to OS in MM patients. Then, a stepwise multivariate Cox regression analysis was performed, and three genes (LYVE1, RNASE1, and RNASE2) were eventually selected to construct a prognostic model (Fig. 6). Risk score = (−0.59714 * expression level of LYVE1) + (−1.35971* expression level of RNASE1) + (0.28240 * expression level of RNASE2). Next, we calculated the optimal cut-off value for the three-gene expression risk score with X-Tile software, and divided patients in training set (n = 241) into high risk group and low risk group. The K–M curve showed that high risk group had poorer prognosis than low risk group (P < 0.0001) (Fig. 7). The AUCs of time-dependent ROC curves were calculated to assess the prognostic capacity of the three-gene signature. The AUCs of risk scores in training set at 8-,10-, and 12-year survival times were 0.717, 0. 720 and 0.800 (Fig. 8A).

Figure 5 Key genes identified from both the DEGs and the hub genes.

A Venn diagram of the DEGs and hub genes revealed 47 key genes.

Figure 6 Relationship of the three prognostic genes with OS.

Kaplan–Meier curve showing that the three genes (LYVE1, RNASE1, and RNASE2) expression were associated with OS of MM patients (A–C).

Figure 7 Construction and validation of prognostic risk model for MM patients.

(A–C) Heatmaps of the three prognostic genes in training set, test set, and whole set, respectively. (D–F) Risk score distributions in training set, test set, and whole set, respectively. (G–I) Kaplan–Meier curve showing that OS is significantly shorter for patients in high-risk group than those in low-risk group in training set, test set, and whole set, respectively.

Figure 8 Time‐dependent ROC curves for the prognostic model in the GEO MM cohort.

(A–C) Time‐dependent ROC curve analysis showing the accuracy and reliability of the prognostic model in training set, test set, and whole set, respectively. (D–F) The ROC curves for 8-, 10-, and 12-year overall survival predictions for the risk model in compare with age, gender, ISS, and R-ISS models in whole set.

To verify the predictive value of the three-gene signature, we used the internal test set (n = 160) and whole set (n = 401) to assess the results from the training set. Consistent of the findings of the training set, the K–M survival curves of test set and whole set revealed that worse OS existed in the high risk groups (Fig. 7). The AUCs of risk scores in test set at 8-,10-, and 12-year survival times were 0.709, 0.768 and 0.766 (Fig. 8B), and the whole set were 0.714, 0.741 and 0.783, respectively (Fig. 8C). We also compared the prognostic capacity of our model with other conventional models, including age, gender, ISS, and R-ISS stage in whole set. The AUCs for 8-, 10-, and 12-year OS predictions for age were 0.500, 0.518, and 0.478; and for gender were 0.516, 0.506, and 0.450 (Figs. 8D–8F). The AUCs of ISS stage at 8-, 10-, and 12-year survival times were 0.503, 0.547, and 0.633; and of R-ISS stage were 0.613, 0.616, and 0.634 (Figs. 8D–8F). Our model had the highest AUCs, indicating a superior prognostic value. Together, these results demonstrated that our model was capable of predicting OS of MM patients.

The prognostic model is independent for MM patients

The univariate and multivariate Cox regression analyses were utilized to investigate independent prognostic parameters and identify whether the three-gene prognostic model was the independent prognostic parameter in 401 MM patients with complete clinical information from GSE136337. Univariate Cox regression analysis showed that NOT, tumor stage (ISS and R-ISS), t(11,14), albumin level, β2-microglobulin level, and the prognostic model had prognostic values (Fig. 9A). After the multivariate Cox regression analysis, NOT, R-ISS, t(11,14), LDH level and the prognostic model were proved to be independent prognostic factors for OS of MM patients (Fig. 9B).

Figure 9 Prognostic risk model could predict prognosis independently.

(A). Univariate association of the prognostic model and clinical characteristics with OS. (B). Multivariate association of the prognostic model and clinical characteristics with OS.

Correlation between the prognostic model with clinical characters

Correlation between risk score and various clinical parameters were shown in Fig. 10. Our results indicated that risk score was significantly correlated with ISS, and R-ISS stage; patients in phase III of ISS or R-ISS had the highest risk score when compared with patients in phase I and II (Fig. 10). However, no differences were found between risk score and age or gender or NTO (Table 1). Besides, the relationships between the three genes in the prognostic model with clinical characters were also analyzed. We found that as LYVE1 expression, which was higher in males than in females, decreased, ISS stage in MM patients increased significantly (Fig. 10A, Table 1). Additionally, as LYVE1 and RNASE1 expression decreased, R-ISS stage in MM patients increased (Figs. 10B and 10C). These findings showed that LYVE1 and RNASE1 exerted protective effects against disease progression.

Figure 10 Relationship of prognostic genes and risk model with clinical variables for MM.

(A and B) LYVE1 expression was correlated with ISS and R-ISS stage in MM patients. (C) RNASE1 expression was correlated with R-ISS stage. (D and E) Risk score was significantly correlated with ISS and R-ISS stage.

Table 1 Correlation analysis between three genes and clinical characters for MM.

Variables	Age (≤60, >60) t (p)	Gender (female, male) t (p)	NTO (≤1, >1) t (p)	ISS (I, II, III) t (p)	RISS (I, II, III) t (p)	
LYVE1	236.601	−2.477*	3.895	9.293*	8.888*	
RNASE1	208.63	−1.529	5.652	5.13	10.782**	
RNASE2	239.084	−2.524*	7.929	0.167	3.094	
Risk score	238.985	−0.208	5.607	6.887*	21.262***	
Notes:

* p < 0.05.

** p < 0.01.

*** p < 0.001.

t: t value from Student’s t test; p: p-value from Student’s test.

Discussion

WGCNA is an important and helpful method which describes the pattern of gene association between different samples, and usually used for gene prediction in large-scale datasets (Zhang & Horvath, 2005; Langfelder, 2008). The main advantage of this method is that it can cluster genes into co-expression modules, as well as build a bridge between gene expression changes and sample characteristics. Thus, WGCNA has been widely used in various malignant and non-malignant diseases (Chen et al., 2020; Chen et al., 2019; Chen et al., 2018; Li et al., 2020a; Zheng et al., 2020). In cancer research, more and more researchers have applied WGCNA for developing new prognostic biomarkers (Xu et al., 2020; Chen et al., 2020; Chen et al., 2019; Chen et al., 2018). In this study, we firstly used an integrated analysis from both WGCNA and DEGs to screen potential biomarkers related to prognosis of MM in GEO databases.

Totally, 178 DEGs were identified, including 59 up-regulated genes and 119 down-regulated genes in GSE6477. Next, we applied WGCNA to construct a co-expression network for evaluating the relationships between genes and modules. In WGCNA, genes were clustered into 15 modules, and the blackmodule and purple module were identified to be the most relevant modules with cancer traits. According to the cut-off criteria: module membership values >0.80 and GS values >0.2, a total of 92 hub genes from these two modules were selected. Key genes were excavated from both the hub genes and the DEGs. Totally, 47 key genes were identified and selected for survival analysis. Using univariate and multivariate Cox regression, we screened three independent prognostic genes and constructed a predictive model. The prognostic genes were LYVE1, RNASE1, and RNASE2. The patients were divided into high- and low-risk groups based on the risk score. Training set, test set and whole set were utilized to validate the prognostic capacity of the our model. Besides, the three-gene prognostic model revealed independence and accuracy in relation to other clinical parameters for MM patients. Our model was the first prognostic model for survival times of MM via integrative bioinformatics analysis.

All of the genes in the three-gene signature have not been previously reported to be associated with MM. LYVE1 is a hyaluronic acid receptor. Its main function is to transport hyaluronidase in extracellular matrix to lymphatic fluid. Also, it involves in the generation of lymphatic vessels and the activation of lymphocytes (Arimoto et al., 2018). High expression level of LYVE1 in tumor tissue can induce tumor cell proliferation, lymphangiogenesis and lymphatic metastasis, and was reported to be closely related to poor prognosis in many cancers, including lung cancer, renal cancer, and breast cancer (Li et al., 2018; Schraml et al., 2019; Hunter et al., 2019). However, LYVE1 might also play anti-tumor role in some cancers. In papillary thyroid carcinoma, LYVE1 was proved to be decreased and associated with poor prognosis (Wu et al., 2019). Latil et al. (2003) reported that LYVE1 was downregulated in prostate cancer and was related to the relapse. In hepatocellular carcinoma and ovarian cancer, its expression was also decreased and played the tumor-suppressive effect (Llovet et al., 2006; Gao et al., 2017). However, its expression and role in MM has not been reported. In this study, we found that the downregulation of LYVE1 in MM was associated with worse prognosis. In addition, the expression level of LYVE1 was closely related to clinical stage (ISS and R-ISS).

RNASE1, belongs to the ribonuclease (RNase) A gene superfamily, acts to regulate the vascular homeostasis of extracellular RNA (Bedenbender et al., 2019). In gastric cancer, RNASE1 acted as a tumor suppressor gene, as its expression was demonstrated to decrease progressively from the normal, primary cancer and metastatic cells (Wang et al., 2006). While in prostate cancer, overexpression of RNASE1 was associated with the poor survival (Gao et al., 2020).

RNASE2, another member of the RNase A gene family, mainly involves in immune function and plays an important role in toll-like receptor activation (Ostendorf et al., 2020). Studies have identified that combination analysis of RNASE2 with other genes has independent prognostic value in kidney renal clear cell carcinoma (Xiang et al., 2020) and renal cell carcinoma (Qin et al., 2021). It has also been found to be upregulated in acute lymphoblastic leukemia (Niini et al., 2002). The upregulation of RNASE2 in gastric cancer was associated with poorer survival (Wang et al., 2020). However, its role in MM is unknown. In our study, overexpressed RNASE2 was found to be associated with worse prognosis in MM patients. The tumor suppressor effect of RNASE2 in MM and its molecular mechanisms need further study.

There are several advantages and limitations in this study. The GEO databases have enough sample size and provide a relatively believable basis for bioinformatics analysis. Besides, in this study, we used an integrated analysis from both WGCNA and DEGs to screen potential biomarkers, which may also make our results more credible. Although our results were validated in other GEO databases, it would be more reliable to validate the significance of prognostic model in real-world clinical MM cohorts. Moreover, biological functions and underlying molecular mechanisms of these key genes in MM are still needed to be explored in the future research.

Conclusions

In summary, we used comprehensive study including WGCNA, DEGs screening, univariate and multivariate Cox regression to construct a three-gene-based prognostic model, which exhibits an effective prognostic value for distinguishing MM patients with poor OS.

Additional Information and Declarations

Competing Interests

Author Contributions

Data Availability

The authors declare that they have no competing interests.

Ying Pan performed the experiments, analyzed the data, prepared figures and/or tables, and approved the final draft.

Ye Meng performed the experiments, analyzed the data, prepared figures and/or tables, and approved the final draft.

Zhimin Zhai conceived and designed the experiments, authored or reviewed drafts of the paper, and approved the final draft.

Shudao Xiong conceived and designed the experiments, authored or reviewed drafts of the paper, and approved the final draft.

The following information was supplied regarding data availability:

The raw data of GSE6477 and GSE136337 were downloaded from the gene expression omnibus (GEO) website. Series matrix were downloaded in GSE136337 for validation.

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
