# Peer review of "Identification of a three-gene-based prognostic model in multiple myeloma using bioinformatics analysis"

_PeerJ, doi:10.7717/peerj.11320_

## Round 0.1 · original submission · Major Revisions

Please revise your manuscript according to the reviewers' comments.

·

Basic reporting

The manuscript is overall well written. However, I would suggest to do one more proofreading to make sure there are no error such as missing pronouns or determiners, such as in line 62 that should be changed to “which was based”. The structure of the article is professional. Regarding references, I would suggest the authors make references for all the programs, software, and packages, as well as their versions and parameters used. In addition, the introduction should discuss the current state of prognosis models based on RNA-seq or single cell RNA-seq data for MM.

Experimental design

- Overall, the study should compare their model results to the ones obtained by the prognosis models used today
- The figures are low quality. Not only in terms of resolution but also in their content. For example: Fig 1a should have a better legend, the labels on Fig 1b are hard to read and no title is provided in the legend and spell out what N and T is, on the network shown on Fig 2 is impossible to see the network structure (it should also show the type of interaction from STRING: direct, predicted, etc). Please make the figure readable and that show what you are trying to show
- Figure 1c shows pathways related to cancer, but how do the authors explain the KEGG results showing a good amount of non-cancer related results?
- Line 110-111: Authors should make the distinction between correlation and anticorrelation. Module purple is not highly correlated with cancer, its anticorrelated.
- Line 115-122: P-values should be adjusted by all comparisons performed. In addition, the hazard ratio/risk ratio and 95% confidence interval should be reported (should also be applied for results associated with fig 7)
- Line 123-127: Can a prognostic model be constructed based on the expression of the final set of prognostic associated genes and show its performance metrics? What about confounding variables that could explain the difference in the high/low risk groups such as sex, age, tumor characteristics, race/socioeconomics, etc?
- How can be confident the prognostic associated set can be applied to other patient cohorts? Not all selected genes for in fig 6 showed prognostic value in fig 7. What happens if another cohort is included? Will only a subset of fig 6 work in it? It is important to note that the prognostic markers (and model that should be constructed after it) is based only in one cohort and didn’t fully “worked” in the validation cohort. The authors should show that their final prognostic model and gene set is not biased because of the cohort selected to construct it and can be applied to real world patient cohorts. Perhaps the authors could construct multiple models based on N differentially expressed or WGCNA hub genes, where N is the number of genes used for the prognostic model, and show that it outperforms these other genes and gene sets from other prognostic models from the literature.

Validity of the findings

It is difficult to assess importance and the validity of the findings in this manuscript given the lack of comparisons with current used methods. In addition, the authors should perform further experiments as suggested in the previous section.

Additional comments

No further comments

Reviewer 2 ·

Basic reporting

The paper of Pan Y. et al. reports on a bio-informatic analysis of Multiple Myeloma (MM) patients’ gene expression data stored in GEO public repository. The paper is clear and formally conform to PeerJ standard; English language is correct. Figures are explicative and clear. The biological context is clearly illustrated in the Introduction, even though more details might be provided on MM prognostication; in fact, recently several reports deeply explored this field, by comprehensive analyses of patients’ genomic profile, and several prognostic scores have been proposed and/or are employed in the management o f MM patients.

Experimental design

The experimental design of the study is clear and rigorous, the main originality being the use of novel bio-informatic approaches leading to the definition of co-expressed modules of genes in MM patients, as compared to healthy donors. The study is aimed at the definition of novel prognostic markers in MM, a quite modern issue, which has been already explored by several authors, by genomic and bio-informatic approaches. MM plasma cells gene expression profile represents one of the features which have been thoroughly analyzed to this aim.
In the present paper, the use of novel bio-informatic tools highlights the expression of a small sub-set of co-expressed genes, whose up-regulations favorably impact patients’ prognosis.
The methods employed are clearly described and enough informative to be replicated. However, no information on disease stage, or therapeutic treatment or patients’ genomic profile (cytogenetic aberrations) have been provided, nor have been considered in the survival analysis.

Validity of the findings

The overall findings of the study are intriguing, since a list of differentially expressed genes is provided, which has been rigorously defined and might be useful to understand MM clinical behavior. However, the prognostic significance of the deregulated expression of these genes need to be more in details analyzed, for instance by multivariate analysis, considering also all the other already well-defined MM prognostic factors (for instance del(17p) and or International Staging System).

Additional comments

The paper is overall interesting, even though it might be improved with a deeper contextualization of the rationale concerning MM prognostic factors and by a more accurate survival analysis, including factors already known to impact on MM patients’ prognosis.

---

## Round 0.2 · accepted · Accept

The authors have answered the questions asked by the reviewers and improved the manuscript, so I suggest it can be accepted for publication.

·

Basic reporting

The authors have improved the manuscript accordingly

Experimental design

The authors have improved the manuscript accordingly

Validity of the findings

The authors have improved the manuscript accordingly